**Data Availability Statement:** Code for reproducing the experiment is available at https://doi.org/10.5281/zenodo.6472931.

**Funding:** The author(s) received no specific funding for this work.

# Environmental effects on the spread of the Neolithic crop package to South Asia

**Jonas Gregorio de Souza**[1]*, **Javier Ruiz-Pérez**[1], **Carla Lancelotti**[1,2], **Marco Madella**[1,2,3]*

1 Department of Humanities, Culture and Socio-Ecological Dynamics Group (CaSEs), Universitat Pompeu Fabra, Barcelona, Spain, 2 Institució Catalana de Recerca i Estudis Avançats (ICREA), Barcelona, Spain, 3 School of Geography, Archaeology and Environmental Studies, The University of the Witwatersrand, Johannesburg, South Africa

* jonas.gregorio@upf.edu (JGS); marco.madella@upf.edu (MM)

## Abstract

The emergence of Neolithic economies and their spread through Eurasia was one of the most crucial transitions of the Holocene, with different mechanisms of diffusion—demic, cultural—being proposed. While this phenomenon has been exhaustively studied in Europe, with repeated attempts to model the speed of Neolithic diffusion based on radiocarbon dates, much less attention has been devoted to the dispersal towards the East, and in particular to South Asia. The Neolithic in the latter region at least partly derived from southwest Asia, given the presence of "founder crops" such as wheat and barley. The process of their eastward diffusion, however, may have been significantly different to the westward dispersal, which was mainly due to demic diffusion, as local domesticates were already available and farming was already practiced in parts of South Asia. Here, we use radiocarbon dates specifically related to the spread of the southwest Asian Neolithic crops to model the speed of dispersal of this agricultural package towards South Asia. To assess potential geographical and environmental effects on the dispersal, we simulate different speeds depending on the biomes being crossed, employing a genetic algorithm to search for the values that most closely approach the radiocarbon dates. We find that the most important barrier to be crossed were the Zagros mountains, where the speed was lowest, possibly due to topography and climate. A large portion of the study area is dominated by deserts and shrublands, where the speed of advance, albeit closer to the range expected for demic diffusion, was lower than observed in Europe, which can also potentially be attributed to environmental constraints in the adaptation of the crops. Finally, a notable acceleration begins in the Indus valley, exceeding the range of demic diffusion in the tropical and subtropical environments east of the Indus. We propose that the latter is due to the rapid diffusion among populations already familiar with plant cultivation.

**Competing interests:** The authors have declared that no competing interests exist.

## 1. Introduction

The origin of agriculture represents a key economic transition in human societies as the cultivation of plants (and partially the herding of animals) allowed for, at least potentially, a much increased rate of population growth [1]. The appearance of this new technology and its spread has been discussed and modeled by several authors, starting with the "wave of advance" proposed by Ammerman and Cavalli-Sforza [2]. This model explained the diffusion of the Neolithic in Europe as a demographic advantage of agriculturalist societies over hunter-gatherers (demic diffusion: see also [3, 4]). Similar approaches have been applied to understand the Neolithisation process in South Asia [5, 6], but much less discussion focused on the dynamics of acquisition of new crops once agriculture was established in South Asia.

South Asia is a region where the transition to agriculture still has many open questions, with potentially both local domestication processes and adoption of exotic domesticates [7]. At the same time, data for the first transition to agriculture is not very common in this part of the world, probably due to taphonomic issues of preservation but also, and more importantly, to the low level of sedentarisation of early agro-pastoral groups (e.g. [8]). Indeed, the deep seasonal variability typical of many areas of the subcontinent would have make sedentary life challenging. Most archaeological evidence, and especially archaeobotanical evidence, comes from more settled (and therefore more visible) archaeological sites of later periods [7]. Therefore, our current knowledge of the process of agricultural developments is more detailed for later periods and settled communities. The earliest archaeological evidence for agriculture in the area comes from aceramic settlements of the highlands west of the Indus River, such as the village of Mehrgarh. This small village, dated to the seventh millennium BCE, is characterised by the presence of domesticated barley (for which there is evidence of local domestication; see [9, 10]) and wheat varieties of southwest Asian origin, as well as sheep and goats [10, 11]. Mehrgarh remains the only excavated evidence for such early dates associated to agricultural economy in the area. From ca. 4000 BCE there is evidence from other early villages in Baluchistan and Khybar Pakhtunkhwa region, which are often positioned on alluvial fans similarly to many West Asia early agricultural settlements [12]. These are ideal settings for practicing farming, which relies on winter or summer rainfall in drylands, especially if water from run-off and from the streams is trapped and channeled in the fields [13]. Through such practices, soils would also improve thanks to the minerals and silt/clay deposited during the periods of inundation.

Such dynamics seems to suggest that there might have been a dispersal from the west of small groups practicing agriculture that were familiar with such ecological settings and preferentially used them for settling. During the fourth millennium BCE there is clear evidence that the agro-pastoral groups of the western flanks of the Indus flood plain spread into different parts of the hills and piedmont area as well as into the main Indus plain. However, it is important to highlight that the very active Indus flood plain implies that earlier villages or camps in this geomorphological region are several meters below the current alluvial deposits and affected by the water table, which makes their identification and excavation very difficult. When we look at the Ganges plain, there is strong evidence for a local Neolithic tradition based on rice cultivation, although some discussion remains if sedentism and villages are a local development or this happened only after contact with groups from the greater Indus [14]. Finally, in south India there is the case for a genuinely independent origin of agriculture with dates significantly later than in the north and north-west (ca. 3000 BCE). The earliest villages (e.g. Kodekal, Utnur or Watgal) have occupation levels but not ashmounds, the accumulations of burnt dung that become the key identifier of the southern Neolithic. Differently from other agricultural traditions, the south India Neolithic is characterised by pastoralism.

This paper presents the results of modeling radiocarbon dates for the South Asian Neolithic, specifically the dates related to the acquisition of new crops coming from Southwest Asia. The rate of spread of the Neolithic from the Near East to Europe, as inferred from radiocarbon dates, has been repeatedly modelled using a variety of approaches such as regression analysis, simulations and cost-distance methods [15–19]. In general, a front speed of ca. 1 km yr$^{-1}$ is estimated, but variations according to terrain (e.g. the effect of sea-voyaging accelerating the dispersal along the Mediterranean coast) and mode of diffusion (e.g. slowest in cases of demic-diffusion, faster in cases of cultural diffusion; see [20, 21]) have been observed. Contrary to the European case, the expansion of the Fertile Crescent agricultural technology towards the East has not received the same attention, with the exception of an exploration of the spread up to the Indus valley by Gangal et al. [6], where it is hypothesised that the Indus Civilization agricultural roots lie in southwest Asia. However, the scarcity of Neolithic evidence and the mixing of traditions, with the acquisition of domesticates from southwest Asia, East Asia and Africa, makes it difficult to disentangle the development of the local agricultural communities, and where farming was the result of immigration, adoption or local domestications. It is in any case clear that the sub-continent has seen patterns of agricultural origins, based on local domestication of wild species such as millets (*Brachiaria ramosa*, *Echinochloa frumentacea*, *Panicum sumatrense*, *Paspalum scrobiculatum*, and *Setaria pumila*), pulses (*Cajanus cajan*, *Macrotyloma uniflorum*, *Vigna mungo*, *Vigna radiata*, and *Vigna aconitifolia*), sesame (*Sesamum indicum*) and cotton (*Gossypium arboreum*).

In this context, it is important to explore the eastward dispersal of the southwest Asia founder crops, especially wheat and barley, to provide a comparison with the dispersal to Europe. The South Asian environment, specifically India's subtropical monsoon climate, offered a challenge to the adaptation of the Neolithic founder crops. The European Mediterranean and temperate climates, on the contrary, are closer to the Levant's, facilitating the adaptation of the Near Eastern crops. Furthermore, unlike Europe, where the Neolithic economy expanded over local Mesolithic economies (by a mix of demic diffusion and admixture; see [20]), local agricultural economies had already emerged in South Asia by the time the Near Eastern package arrived [7].

Here, we focus on the spread of the southwest Asian founder crops, and more specifically the cereals wheat and barley. Wild cereals, including the ancestors of wheat and barley, were processed in the Levant since the Epipaleolithic, ca. 20000 BP [22]. These cereals were fully domesticated by the Pre-Pottery Neolithic B (PPNB) and their dispersal in all directions can be traced to at least ca. 10000–8000 BP, when they reached Greece and the Iranian plateau [7]. Although a polyphyletic origin for domesticated barley is possible, with domestication centres in the Fertile Crescent and the Iranian Plateau (including a suggestion that barley might have been domesticated also at Merghar; see [23]), emmer and einkorn wheat most likely originate from southeast Turkey, in the Karaca Dağ range, although emmer may have a second domestication event in the southern Levant [24]. Other founder crops originate from the Fertile Crescent, and the first archaeobotanical evidence for fully domesticated crops is attested in PPN sites in Turkey, Syria and northern Iraq (e.g. Çayönü, Tell Aswad, Jarmo) [24, 25]. From this cradle, the grain crops eventually expanded over 4000 km to southern India—through what is suggested to be a mixed process of demic diffusion and adoption by local populations (including groups already practicing agropastoralism) [7].

Although the beginnings of this process are most probably linked to demic diffusion, the situation is more complicated further east, where adoption by populations already practicing agriculture seems to be the main mechanism of dispersal of the Near Eastern cereals. For example, in Southern India, a local Neolithic horizon represented by the Ashmound Tradition is dated to at least ca. 5000 BP. However, the earliest phase of the Ashmound Tradition is

characterised by the cultivation of native pulses (mungbean and horsegram). It is only later that the Near Eastern cereals become part of the southern Indian Neolithic economy, together with millets of African origin [7]. Thus, contrary to the European case (where the transition to agriculture is usually assumed to be related to diffusion from the Near East), estimates of the rate of spread of the southwest Asian Neolithic package in South Asia are confounded by the combination of local domestication and adoption of exotic domesticates.

Here, we model the diffusion of the Neolithic package from the Fertile Crescent to South Asia. However, to adopt a similar approach to previous analyses of the Neolithic expansion to Europe, we must explicitly separate the diffusion of the Near Eastern package from the processes of local domestication. At the same time, to take into consideration the variations in the expansion/adoption of that package due to environmental constraints and local processes, we divided the region of interest (from the eastern Fertile Crescent to India) into zones of ecological and cultural significance, estimating separate speeds of advance for each of them. We expect that the southwest Asian crops would have spread with different rhythms depending on environmental constraints and mode of diffusion (demic or cultural), especially east of the Indus valley.

## 2. Materials and methods

### 2.1 Data selection

To specifically address the diffusion of the southwest Asian Neolithic package, we compiled a database of radiocarbon-dated sites affiliated with the spread of such domestic assemblage. In addition to the inclusion of direct dates on wheat, barley, and other plants of Near Eastern origin, or of cultural contexts where such plant remains were found, we decided to include dates from archaeological cultures with archaeobotanical evidence in the literature for the cultivation of those crops [14, 26]. The final dataset contains 143 dates (Fig 1, S1 Table).

In summary, the following criteria have been followed for including dates in the analysis:

- Dates obtained directly from plant remains, such as charred barley or wheat seeds. These constitute the best evidence for the crops of Near Eastern origin.

- Dates on charcoal, ash, bone collagen or other material if they are associated with archaeological cultures or phases known to have cultivated the Near Eastern crops (based on archaeobotanical evidence from other sites).

In the Indus basin, the diffusion of the southwest Asian crops can be chronologically situated during the Indus Valley Civilisation (ca. 5000–4000 BP), with widespread occurrence of wheat and barley in the archaeobotanical record supporting the importance of those cereals in the Indus agriculture [26, 27]. Therefore, we have included dates from sites of the Early and Mature Harappan phases of the Indus Valley Civilisation. Together with pulses of southwest Asian origin, wheat and barley form a package that diffused out of the Greater Indus Valley to east and south India in post-Harappan times [26]. It is important to note that not everywhere in the Harappan domain these crops had the same importance; in Gujarat, for example, they seem to be poorly represented. In northwest India, the archaeobotanical record of Mature Harappan sites attests the ubiquity of barley, the presence of wheat and the mixing of these crops with native millets [28]. Sites of the Kot Diji phase (ca. 5300–4600 BP) were included on the basis of phytolith evidence for wheat and barley at the Kot Diji site [27], as well as on unpublished macro-botanical data from Bhando Qubo, a settlement in the main Indus plain. We stress that the selection of the Indus Valley Civilisation as the moment of diffusion of the southwest Asian winter crops is a conservative estimate, given the scarcity of archaeobotanical evidence from the Indus valley core, and may be revised in the future.

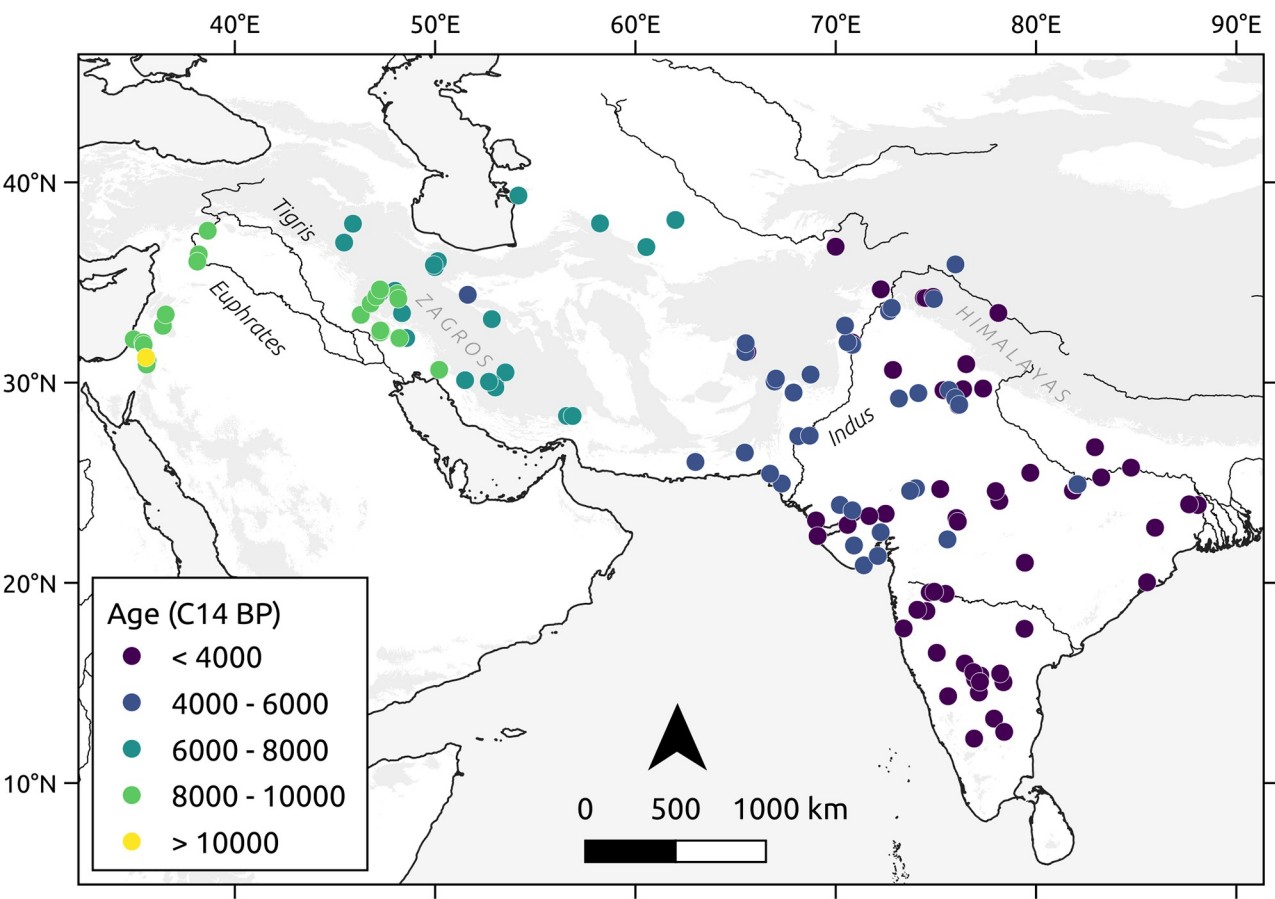

**Fig 1. Dated sites related to the spread of the southwest Asian agricultural package selected for this study.** Grey areas represent elevated regions with > 1000 masl.

Sites belonging to the Chalcolithic cultures of the northern Deccan, in India, mainly the Jorwe and Malwa phases (ca. 3900–2900 BP), have been included based on the archaeobotanical evidence of wheat and barley [29]. These cereals are the most widespread and frequent crops in the archaeobotanical record of Chalcolithic Maharashtra, and form a package with Near Eastern winter pulses, such as lentils and peas [26, 30]. As exemplified by the site of Balathal, wheat and barley appear associated with tropical Indian summer crops—indigenous pulses and small millets [31].

Neolithic sites from the Deccan plateau of southern India (Karnataka and Andhra Pradesh) have the earliest evidence of pastoralism, represented by the beginnings of the Ashmound Tradition [32]. Sites can be divided into an earlier period, when only local domesticates (millets and pulses) are present alongside cattle, sheep and goats, and a later period when wheat and barley are introduced, also marked by a change in ceramic forms [7, 30, 32]. We have included only sites belonging to the Neolithic IIA and later phases (after ca. 4200 BP), as per Fuller et al. [32], which have plenty of archaeobotanical evidence attesting the presence of wheat and barley alongside native crops. The Near Eastern cereals, however, are present in small quantities, and not in all sites [30]. Differently from the northern Deccan, the basis of the southern Neolithic were the indigenous pulses and millets. The phenomenon of the southern Indian Neolithic therefore must be understood as a process of interaction between local agriculture and

**Table 1. Dates with a potential Neolithic association that have been excluded from the analysis.**

| Site | Country | Earliest date (RCYBP) | Reference | Reason for exclusion |
|------|---------|----------------------|-----------|---------------------|
| **Aq Kupruk** | Afghanistan | 7220 ± 100 (Hy-428) | [34] | Pastoralism without evidence for the founder crops |
| **Mehrgarh** | Pakistan | 7328 ± 290 (Beta 1407) | [35] | Still isolated |
| **Budihal** | India | 7725 ± 210 (PRL-1532) | [36] | Pastoralism with local plant domestication (Ashmound Tradition) |
| **Hanumantaraopeta** | India | 3167 ± 40 (R-28680/34) | [37] | Pastoralism with local plant domestication (Ashmound Tradition) |
| **Utnur** | India | 4125 ± 150 (BM-54) | [38] | Pastoralism with local plant domestication (Ashmound Tradition) |
| **Watgal** | India | 4350 ± 100 (PRL-1575) | [39] | Pastoralism with local plant domestication (Ashmound Tradition) |
| **Kodekal** | India | 4415 ±105 (TF-748) | [38] | Pastoralism with local plant domestication (Ashmound Tradition) |
| **Bagor** | India | 5240 ± 80 (TF-1011&1012) | [38] | Mesolithic with domesticated animals |
| **Sannarachamma (Sanganakallu)** | India | 5469 ± 35 (R-28680/22) | [40] | Pastoralism with local plant domestication (Ashmound Tradition) |

pastoralism introduced from the north—with a delay in the adoption of the southwest Asian winter crops [7].

Other sites notable for the antiquity of their Neolithic levels which were excluded from the database are the Ak-Kupruk caves in Afghanistan and Mehrgarh in Pakistan. Although Mehrgarh yielded macro-remains and plant impressions of wheat and barley, the site is an outlier in terms of its Neolithic chronology and it is the only current example of such early chronologies in northwest South Asia. It is also possible that the site's early chronology reflects a unique leapfrog event. As for Ak-Kupruk, the earliest Neolithic evidence is based on faunal remains of sheep and goats [33], and therefore the site was excluded, as our focus was on the spread of the southwest Asian crops. Dates that have been excluded from the analysis are summarized in Table 1.

In addition to dates from the sites and cultures mentioned above, we have included PPNA and EPPNB sites from the Fertile Crescent considered in the existing literature as potential centres of origin based on early evidence of cereal cultivation [41].

## 2.2 Simulation

We simulate diffusion from a centre of origin by taking into account geographical effects on local rates of spread, following Russell et al. [42] and Silva and Steele [18]. This method adds anisotropy to the modelling of dispersals by including a friction surface—similar to the calculation of cost surfaces in GIS—to assess variations in the propagation speed in different directions as a function of the habitats being crossed. This approach has been fruitful when applied, for example, to the Neolithic dispersal in Europe, showing an accelerated speed of advance in zones such as the Mediterranean coast, which has been corroborated by other methods [18, 19].

By assigning different values (expressing relative ease of dispersal) for the rate of dispersal through different categories of terrain, the local speed of propagation can be accelerated or slowed down. In our case, we assign a value of 1 when there is no extra cost of propagating through a cell. A cost of 0.5 means that a cell is crossed twice as fast, while a cost of 2 implies slowing down the propagation speed by half. Once relative costs of traversing different categories of terrain are calculated, arrival times can then be modelled based on the (weighted) distance from a centre of origin and a base speed (in our case, 1 km yr$^{-1}$). Least cost distances were calculated using the algorithm implemented in GRASS 7.8 *r.cost* function with the knight's move, which results in more accurate outputs by considering 16 directions instead of the 8-cell neighborhood [43]. Calculations were performed in R for ease of integration with

the genetic algorithm using the *rgrass7* package [44]. The code and data for reproducing the model is available at https://doi.org/10.5281/zenodo.6472931.

For the centre of origin, considering the eastward diffusion that we are attempting to model, we selected the easternmost among the Near Eastern sites with early cereal cultivation in our dataset—Mureybet, in Syria [41]. Since we are simulating the origin of the Neolithic package in southwest Asia, it would be possible to select other sites, or even multiple sites, especially as the emergence of the Neolithic in this area was probably a regional process (e.g. [17]). As a compromise, we ran a separate model with the origin at the earliest site in the data-set—Dhra, in Jordan [41]—with similar results (S3 Table, S1 and S2 Figs).

The friction surface utilised in the model was constructed by assigning costs for crossing the various ecological regions of southwest and south Asia (Fig 2), defined from the current biome classification [45], similar to the approach adopted in the European case study proposed by Silva and Steele [18]. For simplicity, we merged the biomes covering very small areas or containing no sites with the surrounding ones. In addition, because the Indus constitutes an important cultural and ecological frontier, we further subdivided the biome classification to include the Indus Valley as a separate feature.

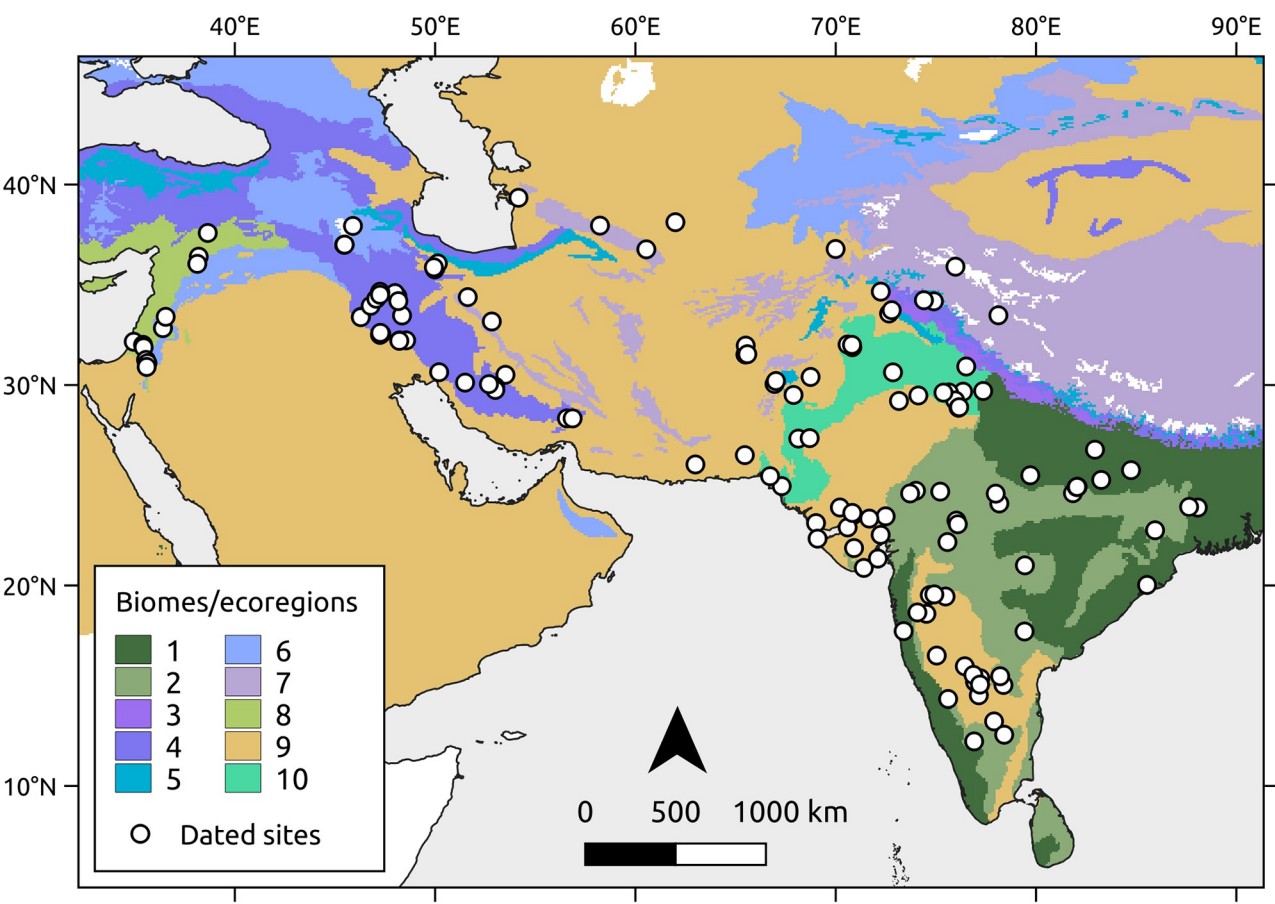

**Fig 2. Biome classification [45] and distribution of dated sites.** 1 = Tropical and subtropical moist broadleaf forests; 2 = Tropical and subtropical dry broadleaf forests; 3 = Tropical and subtropical coniferous forests; 4 = Temperate broadleaf and mixed forests; 5 = Temperate conifer forests; 6 = Temperate grasslands, savannas and shrublands; 7 = Montane grasslands and shrublands; 8 = Mediterranean forests, woodlands, and scrubs; 9 = Deserts and xeric shrublands; 10 = Indus Valley.

Most of the sites in the Fertile Crescent region are distributed across Mediterranean forests, woodlands and scrubs, as well as temperate grasslands, savannas and shrublands. The Zagros mountains, of particular interest for imposing a potential barrier between the Fertile Crescent and the eastern sites, coincides with the temperate broadleaf and mixed forests biome. Across Iran, Afghanistan and Pakistan, sites are found in deserts and xeric shrublands, which dominate most of the study region. Finally, as one reaches the monsoon-driven climate of the Indian subcontinent, sites are distributed across tropical and subtropical moist broadleaf and dry broadleaf forests, but also deserts and xeric shrublands in the southern Deccan plateau.

To evaluate the results of the model, we used the root mean square error (RMSE) between the simulated arrival times and the median of the calibrated 14C dates.

## 2.3 Genetic algorithm

We are interested in finding the cost values (and, consequently, the rate of spread) for each of the terrain classes (biomes) that results in the best fit with the archaeological dates. However, due to the high dimensionality of the parameter space, manually exploring it would be slow if not unfeasible. In such cases, genetic algorithms are commonly used to search the parameter space in order to optimise a given function—in our case, the RMSE between simulated and real dates [18, 42]. Genetic algorithms work by mimicking natural selection: a population of models is initialised with random values for all parameters; the fitness of each model is calculated; the models with the best score are transmitted to the next generation, while the worst performing models are discarded; the best models are allowed to "reproduce", generating new models with crossover (each "parent" model contributes with a subset of its parameter set) and random mutations; the process is repeated for a number of generations until population converges on the optimal parameter set [46, 47].

We start our genetic algorithm with 500 models whose parameters (the relative cost of traversing each terrain class) are randomly initialised from a normal distribution with $\mu = 1$ and $\sigma = 1$. At every iteration, for each model, simulated arrival times are generated, and the fitness score is calculated. The 250 best models (with the lowest error between simulated and archaeological arrival times) are selected to "reproduce" for the next generation, and the 50 best models are preserved without change (a procedure known as elitism). The number of "children" is such that the population size (n = 500) is kept constant. During reproduction, crossover is applied. Mutations happen with a probability of 20%, whereby one of the parameters is changed by adding a value from a standard normal distribution. The relatively high mutation rate helps to avoid the algorithm being stuck in a local optimum by sampling larger areas of the parameter space, while elitism avoids the loss of the best parameter sets to mutation [18, 42]. The algorithm runs for 20 generations, converging early in the process (S3 and S4 Figs).

## 3. Results and discussion

The best parameter set results in a wide range of propagation speeds from the Near East to South Asia. As a general trend, there is an acceleration from west to east, with the slowest speeds observed at the Fertile Crescent and while crossing the Zagros, and the fastest speed being achieved once the Indus is crossed (Table 2, Figs 3 and 4, S2 Table).

A previous study estimated a speed of 0.65 km/yr for the spread of the Neolithic to South Asia, while suggesting that local variations must have existed, e.g. depending on the environments being crossed [6]. The aforementioned study, however, did not include the southernmost Indian sites in the analysis. As noticed by Gangal et al. [6] and supported by our results, the propagation speed (except for tropical India) is slower than the one observed in the Neolithic spread from the Near East to Europe.

**Table 2. Simulated speeds in the main biomes of the study area as selected by the genetic algorithm.**

| Region | Terrain category (biome) | Simulated speed (km yr$^{-1}$) |
|---|---|---|
| **Fertile crescent** | Mediterranean Forests, Woodlands & Scrub | 0.52 |
| **Zagros** | Temperate Broadleaf & Mixed Forests | 0.33 |
| **Iran/Afghanistan/Pakistan** | Deserts & Xeric Shrublands | 0.62 |
| **Indus Valley** | Norhtwestern Thorn Scrub Forests | 1.33 |
| **India** | Tropical & Subtropical Moist Broadleaf Forests | 1.88 |
| **India** | Tropical & Subtropical Dry Broadleaf Forests | 1.76 |

One of the lowest speeds (0.33 km yr$^{-1}$) was observed in the Zagros mountains range. If we assume that the Neolithic spread to the Zagros was due to demic diffusion, the slow speed of expansion—below the range estimated for demic diffusion, ca. 0.7–1.4 km yr$^{-1}$ [16]—can be attributed to the fact that the mountain range acted as a physical barrier. In addition, the adaptation of the crops to the colder environment of the mountain range may have slowed down the expansion. However, there is evidence that the Zagros may have been one of the independent centres of domestication based on the continuous record of management of progenitors of barley, wheat and lentil, as well as early presence of domesticated emmer [48]. Furthermore, ancient genomes from early Neolithic individuals in the region show affinities with modern Iranian, Afghan and Pakistani populations, supporting that an eastward demic expansion would have originated in the Zagros—contrasting with the Anatolian origin of the European farmers

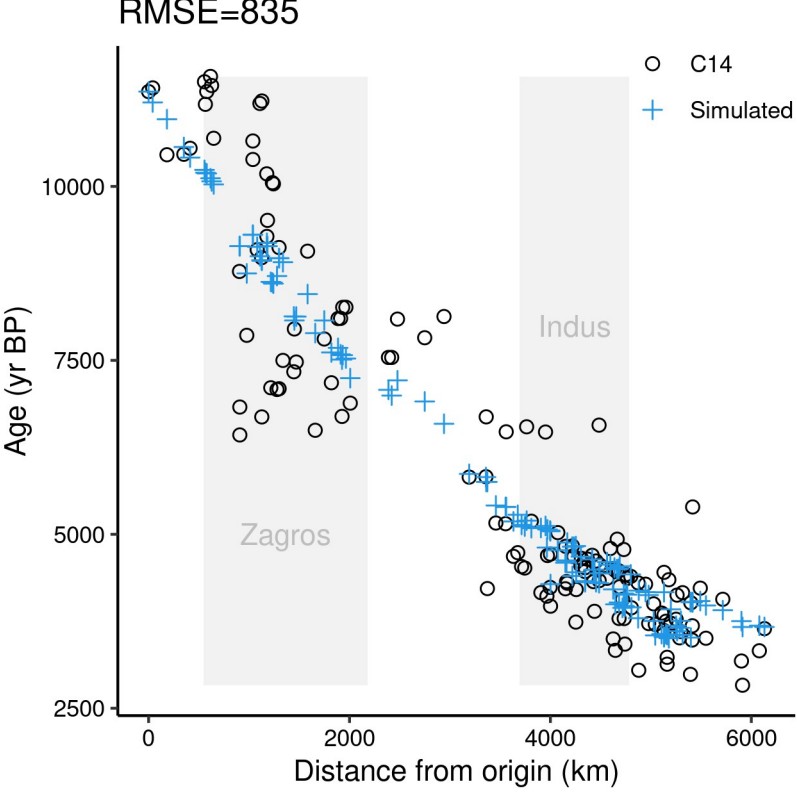

**Fig 3. Comparison between the radiocarbon dates (median of the calibrated distribution) and the simulated arrival times using the optimal parameter set.** The approximate distances to the Zagros Mountains and the Indus Valley are also shown.

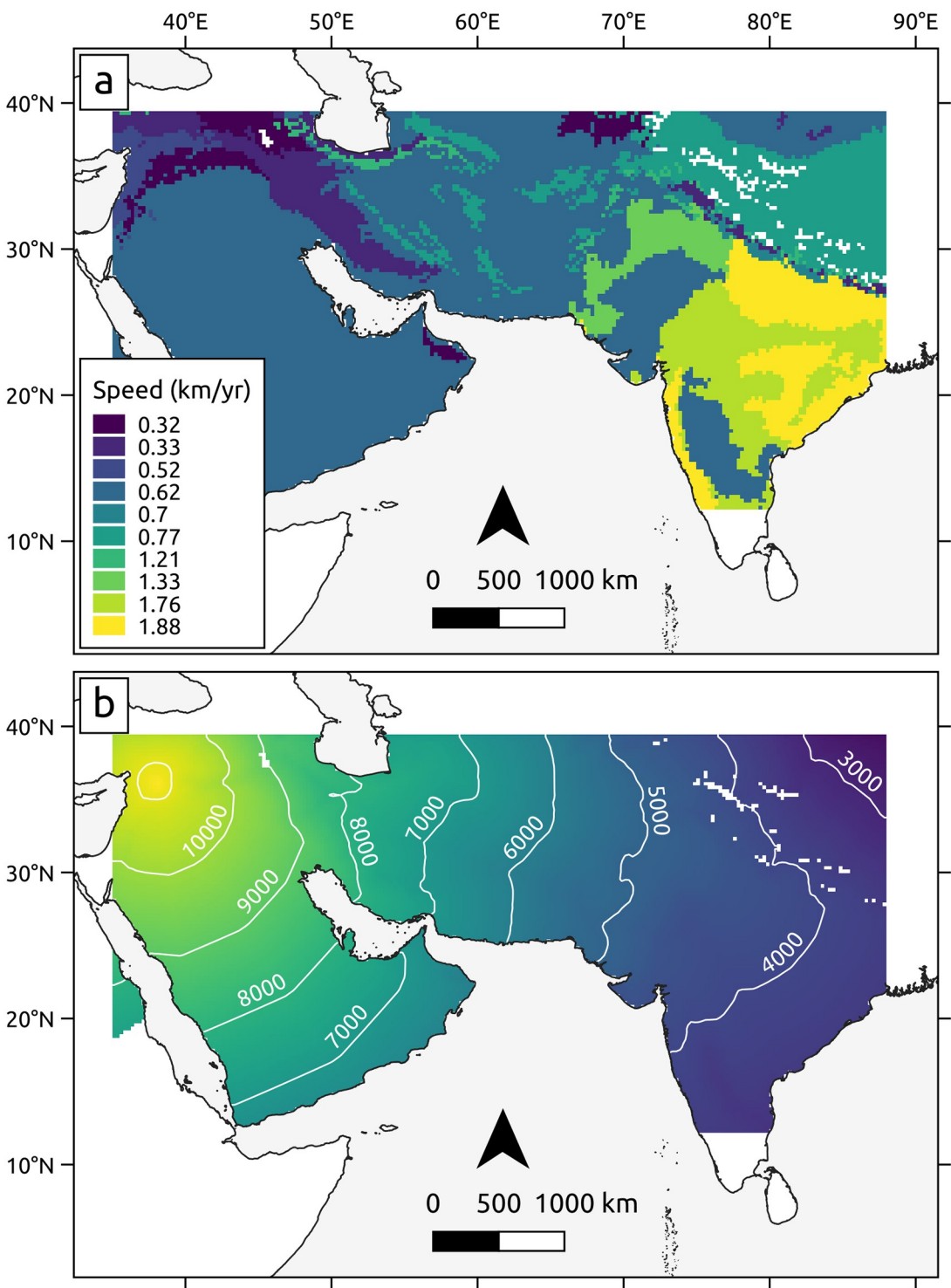

**Fig 4.** a) Simulated speeds of advance for each terrestrial ecoregion according to the optimal parameter set. b) Simulated arrival times with contour lines (yr BP) shown every 1000 years.

[49, 50]. The acceleration east of the Zagros, as observed in our model, may be related to such demic diffusion process, but is still below the range expected for such phenomena, which could be attributed to the terrains being crossed [6]. An ancient genome from a Mature Harappan phase individual from Rakhighari highlighted that there is little if any genetic contribution from steppe pastoralists or western Iranian farmers/herders [51]. This suggests that farmers from the west do not seem to be responsible (or there is little genetic evidence) for the agricultural innovation in South Asia. However, the Indus area might be seen as an area of frontier between western and eastern agricultural "traditions", where some population movement from the west and local developments might have created a more nuanced situation—a transition from moving frontiers (demic diffusion) and static frontiers (interaction between hunter-gatherers and farmers) [7]. Our work suggests that the expansion of the southwest Asia crop assemblage (not of agriculture *per se*) had a population component up to the "frontier" of the Indus.

We suggest that the rate of spread was closer to what is expected of demic diffusion in most of the remaining areas, albeit in the lower end, except for potential geographical barriers such as the Zagros. As proposed by Gangal et al. [6], the drier environment, rougher topography, and absence of a major riverine connection in the W-E axis of the spread may be the main reasons behind the low observed speed in the desert and shrubland biomes that occur in much of the study area.

An acceleration occurs when the Indus Valley is reached, and the highest speeds are observed east of the Indus, when the subtropical and tropical environments of South Asia are reached (1.76–1.88 km $yr^{-1}$). This represents more than a doubling of the speed of advance west of the Indus, and is slightly outside of the range of demic diffusion. If we model the dispersal from the earliest rather than the easternmost Fertile Crescent site, the speed in the Indian subcontinent is even higher, 1.59–3.1 km $yr^{-1}$ (S3 Table, S2 Fig). We consider that the relative acceleration compared to the rates west of the Indus, coupled with the fact that an agro-pastoral lifestyle partly propelled by local domestication was already established in parts of India, suggests that rapid cultural acquisition of the southwest Asian crops was the main mechanism of their dispersal [7].

Wheat can be grown in tropical and sub-tropical areas as a winter crop (*rabi*) and its growth is favoured by cool and moist weather followed by dry and warm conditions to enable grains to ripe properly [52]. The temperature conditions at the time of grain filling and development are very crucial for yield and temperatures above 25˚C during this period tend to depress grain weight. If temperatures are high, plants spend too much energy through transpiration and reducing energy for grain formation resulting in lower yields and quality [52]. Similar conditions can be proposed for barley.

We suggest that, once the adaptive ecological barrier represented by the subtropical monsoon climate was overcome, winter cereals were quickly diffused among the Neolithic populations of the Indian sub-continent, who were already familiar with plant cultivation, resulting in the high observed speed.

## 4. Conclusions

The acquisition of the Near East agricultural package in South Asia, together with the millets of African origins, can be considered one of the first examples of 'food globalisation' in the Old World. Apart from the routes and pace of this process, which have been explored also in the past (e.g. [7]), a set of additional questions were considered in the present study concerning the mechanisms of dispersal and the cultural and environmental context of such spread, as well as the impact of the new crops on the agrarian production and food consumption in the greater Indus Valley and beyond.

A previous analysis of the radiocarbon record related to the Neolithic expansion from the Near East to India calculated an average speed of 0.65 km yr$^{-1}$, lower than that observed for the westward spread towards Europe [6]. While recognizing the simplicity of the model, which considers only the average speed, Gangal et al. [6] notice that local spread rates were variable, and point to environmental effects as a likely explanation for the overall low speed of advance—such as the aridity of most of the territory or the lack of major rivers along the axis of dispersal.

Our results confirm that the eastward advance of the Near Eastern crops proceeded at highly variable speeds. Similar scenarios have been observed elsewhere. For example, the spread of the Neolithic across Europe is known to have followed different paces depending on the ecoregions being crossed—such as the Mediterranean coast, where sea voyaging promoted an acceleration in the front speed [18, 19, 21].

In our analysis of the advance towards South Asia, the main feature observed was a noticeable acceleration in the diffusion rate of the Near Eastern crops once the subtropical environments of India were reached. Assuming that cultural diffusion should proceed at a faster pace than demic diffusion [20, 21], and considering the high speed observed (close to 2 km yr$^{-1}$), one explanation for such acceleration is the fact that, while the Indus acted as a frontier, people inhabiting the areas east of the Indus were already familiar with the cultivation of other plants, including locally domesticated ones—an agricultural complex to which the Near Eastern crops were added as a further element. Future work must concentrate on augmenting the absolute chronology from the Indus Valley sites, resolving local chronologies (for example, outlier dates such as those of Mehrgarh, which were not considered here), obtaining more precise dates, and on developing more sophisticated models, such as agent-based models, to simulate scenarios of demic and cultural diffusion of the Near Eastern crops as well as local processes of emergence of agriculture.

## Supporting information

**S1 Fig. Radiocarbon dates and simulated arrival times (from Dhra).** Comparison between the radiocarbon dates (median of the calibrated distribution) and the simulated arrival times using the optimal parameter set and the site of Dhra, Jordan, as the origin. The approximate distances to the Zagros Mountains and the Indus Valley are also shown.
(TIF)

**S2 Fig. Simulated speeds and arrival times (from Dhra).** a) Simulated speeds of advance for each terrestrial ecoregion according to the optimal parameter set using the site of Dhra as the origin. b) Simulated arrival times from Dhra with contour lines (yr BP) shown every 1000 years.
(TIF)

**S3 Fig. Performance of the genetic algorithm (from Mureybet).** The average and the best (lowest) RMSE between simulated and radiocarbon dates are shown for each generation.
(TIF)

**S4 Fig. Performance of the genetic algorithm (from Dhra).** The average and the best (lowest) RMSE between simulated and radiocarbon dates are shown for each generation.
(TIF)

**S1 Table. Radiocarbon dates.** Radiocarbon dates used in this study.
(CSV)

**S2 Table. Speeds assigned to each biome (from Mureybet).** Speeds assigned to each biome at the end of the genetic algorithm using the site of Mureybet as the origin.
(CSV)

**S3 Table. Speeds assigned to each biome (from Dhra).** Speeds assigned to each biome at the end of the genetic algorithm using the site of Dhra as the origin.
(CSV)

## Author Contributions

**Conceptualization:** Jonas Gregorio de Souza, Javier Ruiz-Pérez, Carla Lancelotti, Marco Madella.

**Data curation:** Jonas Gregorio de Souza, Javier Ruiz-Pérez, Carla Lancelotti.

**Formal analysis:** Jonas Gregorio de Souza.

**Investigation:** Jonas Gregorio de Souza, Javier Ruiz-Pérez, Marco Madella.

**Methodology:** Jonas Gregorio de Souza, Javier Ruiz-Pérez.

**Software:** Jonas Gregorio de Souza.

**Supervision:** Carla Lancelotti, Marco Madella.

**Writing – original draft:** Jonas Gregorio de Souza, Javier Ruiz-Pérez, Marco Madella.

**Writing – review & editing:** Jonas Gregorio de Souza, Javier Ruiz-Pérez, Carla Lancelotti, Marco Madella.

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
