## [Decision Letter · Decision Letter 0]

12 Apr 2022

PONE-D-22-07183Environmental and geographical effects on the spread of the Neolithic crop package to South AsiaPLOS ONE

Dear Dr. Gregorio de Souza,

Thank you for submitting your manuscript to PLOS ONE. After careful consideration, we feel that it has merit but does not fully meet PLOS ONE’s publication criteria as it currently stands. Therefore, we invite you to submit a revised version of the manuscript that addresses the points raised during the review process. This paper received two solidreviews from highly qualified reviewers who both have detailed familiarity with the topic and archaeological and archaeobotanical records of the region. Both reviewers are extremely supportive of the publication of the manuscript. In your revision, I would encourage you to address the minor issues raised by the reviewers.

We look forward to receiving your revised manuscript.

Kind regards,

Xinyi Liu

Academic Editor

PLOS ONE

Journal Requirements:

Reviewers' comments:

Reviewer's Responses to Questions

**Comments to the Author**

1. Is the manuscript technically sound, and do the data support the conclusions?

Reviewer #1: Yes

Reviewer #2: Yes

2. Has the statistical analysis been performed appropriately and rigorously? 

Reviewer #1: I Don't Know

Reviewer #2: Yes

3. Have the authors made all data underlying the findings in their manuscript fully available?

Reviewer #1: Yes

Reviewer #2: Yes

4. Is the manuscript presented in an intelligible fashion and written in standard English?

Reviewer #1: Yes

Reviewer #2: Yes

5. Review Comments to the Author

Reviewer #1: It is a great paper that aims to explain the rates of dispersal of southwestern crop species to southeast Asia. It is a valuable contribution that should be published. I have just a few minor comments

Please clarify the selection of site criteria as a bunch of reference, that do contain directly dated cereal grain, are missing. Eg.previous work in Kashmir:

Betts A, Yatoo M, Spate M et al (2019) The Northern Neolithic of the Western Himalayas: New research in

the Kashmir Valley. Archaeol Res Asia 18:17-39. https://doi.org/10.1016/j.ara.2019.02.001

Yatoo MA, Spate M, Betts A, Pokharia AK, Shah MA (2020) New evidence from the Kashmir Valley

indicates the adoption of East and West Asian crops in the western Himalayas by 4400 years ago. Quat Sci

Adv 2:100011. https://doi.org/10.1016/j.qsa.2020.100011

Lines 31-32 Is “westwards, correct or it should we eastwards?

Lines 53 Please indicate that you mean southeast Asia here, as in China such questions were previous widely discussed, such as wheat and barley joining already established millet or rice agriculture.

Reviewer #2: The manuscript models the timing of the spread of West Asian domesticated crops to South Asia, arguing for a slow movement across the Zagros and Iranian plateau, likely reflecting demic diffusion through a difficult geography, followed by rapid spread of these crops once they reached the Indus frontier and were able to be adapted into indigenous food production systems. The arguments are clear and well supported, taking into account the current archaeological, archaeobotanical, genetic and geographic literature. The authors note the limitations of the study being based on the currently available dates and that new research may change their current hypothesis.

The article is well written, and I see no errors for editing or correction.

I have only one query – there appear a few sites with directly dated crop remains from northern Pakistan/Western Himalayas in the range 6000-4000 BP, e.g. Sheri Khan Tarakai (Khan et al., 2010) and Pethpuran Teng (Yatoo et al., 2020) that were not included in the model in table S1, however I don’t suspect the inclusion of these would significantly change the results of the modelling.

I am happy to recommend the manuscript for publication without revision.

6. PLOS authors have the option to publish the peer review history of their article (what does this mean?). If published, this will include your full peer review and any attached files.

Reviewer #1: No

Reviewer #2: No

---

## [Author Response · Author response to Decision Letter 0]

25 Apr 2022

Reviewer #1: It is a great paper that aims to explain the rates of dispersal of southwestern crop species to southeast Asia. It is a valuable contribution that should be published. I have just a few minor comments

Please clarify the selection of site criteria as a bunch of reference, that do contain directly dated cereal grain, are missing. Eg.previous work in Kashmir:

Betts A, Yatoo M, Spate M et al (2019) The Northern Neolithic of the Western Himalayas: New research in

the Kashmir Valley. Archaeol Res Asia 18:17-39. https://doi.org/10.1016/j.ara.2019.02.001

Yatoo MA, Spate M, Betts A, Pokharia AK, Shah MA (2020) New evidence from the Kashmir Valley

indicates the adoption of East and West Asian crops in the western Himalayas by 4400 years ago. Quat Sci

Adv 2:100011. https://doi.org/10.1016/j.qsa.2020.100011

- We have included dates from the sites figuring in those publications (Pethpuran Teng and Qasim Bagh), as can be seen in Table S1.

Lines 31-32 Is “westwards, correct or it should we eastwards?

- We are referring to the spread towards Europe, in contrast to the phenomenon we are analysing. However, to further clarify that point, we now explicitly mention “eastward diffusion” in the preceding line.

Lines 53 Please indicate that you mean southeast Asia here, as in China such questions were previous widely discussed, such as wheat and barley joining already established millet or rice agriculture.

- We have added “South Asia” to make it clear that we are referring to that region.

Reviewer #2: The manuscript models the timing of the spread of West Asian domesticated crops to South Asia, arguing for a slow movement across the Zagros and Iranian plateau, likely reflecting demic diffusion through a difficult geography, followed by rapid spread of these crops once they reached the Indus frontier and were able to be adapted into indigenous food production systems. The arguments are clear and well supported, taking into account the current archaeological, archaeobotanical, genetic and geographic literature. The authors note the limitations of the study being based on the currently available dates and that new research may change their current hypothesis.

The article is well written, and I see no errors for editing or correction.

I have only one query – there appear a few sites with directly dated crop remains from northern Pakistan/Western Himalayas in the range 6000-4000 BP, e.g. Sheri Khan Tarakai (Khan et al., 2010) and Pethpuran Teng (Yatoo et al., 2020) that were not included in the model in table S1, however I don’t suspect the inclusion of these would significantly change the results of the modelling.

- We have included the date for Pethpuran Teng, as also requested by Reviewer #1, and the date for Sheri Khan Tarakai, as can be seen in Table S1.

I am happy to recommend the manuscript for publication without revision.

---

## [Editor Report · Decision Letter 1]

1 May 2022

Environmental effects on the spread of the Neolithic crop package to South Asia

PONE-D-22-07183R1

Dear Jonas Gregorio de Souza

We’re pleased to inform you that your manuscript has been judged scientifically suitable for publication and will be formally accepted for publication once it meets all outstanding technical requirements.

Kind regards,

Xinyi Liu

Academic Editor

PLOS ONE
---

## [Editor Report · Acceptance letter]

1 Jul 2022

PONE-D-22-07183R1 

Environmental effects on the spread of the Neolithic crop package to South Asia 

Dear Dr. Gregorio de Souza:

I'm pleased to inform you that your manuscript has been deemed suitable for publication in PLOS ONE. Congratulations! Your manuscript is now with our production department. 

Kind regards, 

on behalf of

Dr. Xinyi Liu 

Academic Editor

PLOS ONE